# The Inhibition Property and Mechanism of a Novel Low Molecular Weight Zwitterionic Copolymer for Improving Wellbore Stability

**DOI:** 10.3390/polym12030708

**Published:** 2020-03-23

**Authors:** Weichao Du, Michal Slaný, Xiangyun Wang, Gang Chen, Jie Zhang

**Affiliations:** 1School of Chemistry and Chemical Engineering, Shaanxi Province Key Laboratory of Environmental Pollution Control and Reservoir Protection Technology of Oilfields, Xi’an Shiyou University, Xi’an 710065, China; 2State Key Laboratory of Oil & Gas Reservoir Geology and Exploitation, Chengdu University of Technology, Chengdu 620021, China; 3Institute of Inorganic Chemistry, Slovak Academy of Sciences, Dúbravská cesta 9, 845 36 Bratislava, Slovakia

**Keywords:** low molecular weight, copolymer, shale hydration inhibitor, shale gas, water-based drilling fluids, clay-polymer nanocomposite, montmorillonite

## Abstract

In this work, a novel low molecular weight zwitterionic copolymer for improving wellbore stability, which is expected to be an alternative to the current shale inhibitors, was obtained by copolymerization of tris hydroxyethyl allyl ammonium bromide (THAAB), 2-acrylamido-2- methyl propane sulfonic acid (AMPS) and acrylamide (AM), initiated by a redox initiation system in an aqueous solution. The copolymer, denoted as SX-1, was characterized by FT-IR, TGA-DSC, and GPC. Results demonstrated that the molecular weight of SX-1 was approximately 13,683 g/mol and it displayed temperature resistance up to 225 °C. Regarding the inhibition performance, evaluation experiments showed the hot rolling recovery of a Longmaxi shale sample in 2.0 wt % SX-1 solutions was up to 90.31% after hot rolling for 16 h at 120 °C. The Linear swelling height of Na-MMT artificial core in 2.0 wt % SX-1 solution was just 4.74 mm after 16 h. Methods including particle size analysis, FTIR, XRD, and SEM were utilized to study the inhibition mechanism of SX-1; results demonstrated that SX-1 had entered into the inner layer of sodium montmorillonite (Na-MMT) and adsorbed on the inner surface, and the micro-structure of Na-MMT was successfully changed by SX-1. The particle size of Na-MMT in distilled water was 8.05 μm, and it was observed that its size had increased to 603 μm after the addition of 2.0 wt % of SX-1. Its superior properties make this novel low molecular weight copolymer promising for ensuring wellbore stability, particularly for high temperature wells.

## 1. Introduction

The exploitation of unconventional reservoirs such as shale gas has attracted attention worldwide due to the excessive depletion of conventional oil and gas reserves. However, wellbore instability is the main problem associated with the drilling operation of shale formations, and the main cause of wellbore instability is the hydration expansion in shale formations [1]. The types of drilling fluids are generally divided into water-based drilling fluids, oil-based drilling fluids, and synthetic-based drilling fluids. The drilling fluids play an important role in maintaining the stability of the well wall, cleaning the drill bit, and suspending cuttings. A very important additive in drilling fluids is shale inhibitors, as these play a decisive role in preventing clay hydration and expansion [2]. Generally, oil-based muds (OBMs) are the primary choice to avoid wellbore instability problems due to their superior shale hydration inhibition properties. Unfortunately, environmental restrictions and high costs have largely prevented the wider use of OBMs [3]. At the same time, water-based drilling fluids (WBDFs) shows great potential applications in the exploitation of shale gas due to their simple formulation and low costs. Since wellbore instability is due to the hydration of shale, the development of shale hydration inhibitors and high performance WBDFs has become an important research topic for the oil and gas exploration industry [4].

In recent decades, various compounds, such as simple inorganic monomeric amines, have been successfully employed as shale inhibitors in WBDFs. However, the temporary and low level of shale inhibition has limited their widespread application [5,6,7]. To overcome the performance-related shortcomings, some copolymers such as HPAM (Hydrolyzed polyacrylamide), hyperbranched polyglycerols, FA-367 (a zwitterionic polymer), PDADMAC (Polydimethyl allyl ammonium chloride) and polyamines have been investigated [8]. Researchers have reported studies on acrylamide-based hydrophobic associating polymers as shale inhibitors [9,10], but there are still some shortcomings to be overcome, such as poor temperature resistance. Existing inhibitors include numerous large molecular weight copolymers, which can be wrapped in surface of clay and form a coating film to effectively prevent wellbore instability due to the hydration of shale. However, high molecular weight copolymers also have a great influence on the rheological properties of WBDFs, and high molecular weight copolymer chains are easily twisted and broken in the high-salt and high-calcium drilling fluid environment. Studies have found that polymers with low molecular weight show excellent salt and temperature resistance ability in WBDFs. Low molecular weight polymers such as polyamine, a hyperbranched polymer, have attracted considerable interest from oil-field researchers in the field of shale drill operations in recent years [11]. AMPS copolymer can inhibit fluid loss and be used in oil-field environments as scale inhibitors, friction reducers and water-control polymers. Zwitterionic polymers possess both an anionic and a cationic group, and have great advantages in shale inhibition and maintaining the WBDFs’ performance. Researchers have synthesized a zwitterionic polymer P (AM-DMC-AMPS) as a low-molecular-weight encapsulator in deepwater drilling fluid, and it showed strong shale inhibition performance [12]. So far, however, there are few reports on low molecular weight zwitterionic copolymers as shale inhibitors [13,14,15]. 

Polymer clay nanocomposites have been studied extensively also as a new generation of polymeric materials and received wide interests in the research community of material sciences and engineering due to their exceptional thermal and mechanical properties. Analytical studies suggest that the high clay stiffness, high aspect ratio and volume fraction are the key parameters governing the stiffness of clay nanocomposite. As the mechanical properties of a certain polymer clay nanocomposite system are affected by multiple uncertain parameters such as the clay volume fraction, clay aspect ratio, clay curvature, clay stiffness and epoxy stiffness, a comprehensive study of the effects of such above-mentioned parameters on the mechanical behavior of PCNs is required. The characteristics of the clay particles (their aspect ratio (length), curvature (radius) and stiffness) are randomly generated according to predefined sets of statistical distributions e.g., all clay particles have the same aspect ratio, curvature radius, stiffness, and the clay particles are randomly dispersed in the epoxy matrix. The resulting clay particle configurations are ideal for estimating the influence of each particle-characteristic on the mechanical response [16,17,18,19].

In the present work, AMPS and AM were copolymerized with a novel cationic monomer tris hydroxyethyl allyl ammonium bromide (THAAB) to synthesize a low molecular weight zwitterionic copolymer SX-1 for ensuring wellbore stability. In the polymer, THAAB serves as a functional monomer which ensures the strong adsorption of SX-1 onto the shale surface by reacting with the silanol groups and negative charges on the clay. AM acts as the backbone, and AMPS as a hydration monomer, giving the copolymer outstanding salt and temperature resistance properties. The copolymer was characterized by FT-IR, TGA and GPC. The inhibition properties were evaluated by rolling recovery rate experiments and linear swelling experiments. The inhibition mechanism was analyzed and discussed in detail with the aid of particle size analysis, FT-IR, XRD, and SEM. 

## 2. Experimental

### 2.1. Materials

AMPS, AM, triethanolamine (TEA), allyl bromide, (NH_4_)_2_S_2_O_8_, NaHSO_3_, ethanol, ethyl acetate, KCl and isopropanol were all of analytical purity and purchased from Kelon Chemical Reagent Factory, Chengdu, China. Ultrahib (polyamine) with molecular weight 1352 g/mol was supplied by Maikeba Mud Co., Ltd., Shenzhen, China. SIAT (polyamine) with molecular weight 960 g/mol was received from the Engineering Technology Research Institute Co., Ltd., China National Petroleum Corporation, Beijing, China. Polyacrylamide potassium (KPAM), with molecular weight 1.8 × 10^5^ g/mol, and FA-367, with molecular weight 1.2 × 10^4^ g/mol, were supplied by Sichuan Guangya polymer technology company, Ltd., Chengdu, China. Sodium montmorillonite with cation exchange capacity 82 meq/100 g (0.82 mmol/g) was obtained from Xiazijie Bentonite Technology Co., Ltd., Xinjiang, China. The shale samples were obtained from the Longmaxi shale gas field, Chongqing, China, and the mineral composition is listed in Table 1.

### 2.2. Synthesis of the Polymeric Monomer THAAB

The preparation of THAAB was based on a previously reported method [20,21] shown in Figure 1. Triethanolamine (0.20 mol), allyl bromide (0.20 mol), and ethanol (150 mL) were placed in a round-bottom flask equipped with a reflux condenser and refluxed for 24 h with magnetic stirring at 50 °C. After cooling to room temperature, the solution was concentrated under reduced pressure and redissolved in ethyl acetate and ethanol (ethyl acetate: ethanol ratio 7:3). The product was isolated in high yield as rod-like crystals or white powder after being left at room temperature for 24h. Yield: 94.3%. ^1^H NMR (400 MHz, D_2_O): 5.62–5.75 (m, 1H, CH=C), 4.89~5.07 (m, 2H, C=CH_2_), 3.75~3.76 (d, 6H, CH_2_–O), 3.60 (t, 2H, –CH–C=C), 3.33 (t, 3H, –OH), 3.04~3.06(t, 6H, N–CH_2_–C–OH); IR (KBr), /cm^–1^: 3360, 2920, 1630, 1400, 1080, 900, 520. 

### 2.3. Synthesis of SX-1

SX-1 was synthesized by redox free radical copolymerization in aqueous solution. The synthesis route of SX-1 is shown in Figure 2. Appropriate amounts of AMPS (0.50 g) were dissolved in deionized water (20 mL), and the pH was adjusted to 7.0 by using 30 wt % NaOH solutions. AM and THAAB were then added to the flask with stirring at constant temperature under nitrogen atmosphere for 10 min. Thereafter, the initiator K_2_S_2_O_8_ (0.7 g) and NaHSO_3_ (0.09 g), giving an initiator concentration of 5.0 wt % relative to the total monomer amount, were added. The polymerization was carried out at 90 °C for 0.5 h while stirring, and then isopropanol was added to the solution. Polymerization proceeded at 90 °C for another 1 h. The resulting product was purified by repeatedly washing with ethanol to remove monomers, isopropanol and initiator. SX-1 was then dried in a vacuum oven at 60 °C for 24 h.

### 2.4. Methods

#### 2.4.1. Characterization of SX-1

FT-IR spectra were collected using a WQF-520 Fourier Transform Infrared (FT-IR) spectrometer (Beijing Jingke Ruida Technology Co., Ltd., Beijing, China). The IR source, KBr beam splitter and DTGS detector were used to gather mid-IR measurements (4000–400 cm^−1^). The MIR transmission spectra were collected using the KBr pellet technique (1 mg of a sample homogenized with 200 mg KBr). GPC was utilized to measure the molecular weight using an Alliance e2695 instrument (Waters, MA, USA), the injection volume and operation hours were 50 mL; the testing temperature was 30 °C, flow rate was 1.0 mL /min, standard was polystyrene and solven was tetrahydrofuran. TGA-DSC tests were performed on a simultaneous TGA-DSC (METTLER TOLEDO, Zurich, Switzerland) instrument under nitrogen atmosphere flow (20 mL min^−1^) with a heating rate of 20 °C min^−1^ from 30 to 600 °C. 

#### 2.4.2. Inhibition Performance Evaluations

Linear swelling tests were carried out in the laboratory by using CPZ-2 swelling apparatus (Haitongda Co., Ltd., Qingdao, China). The artificial core was made by using 10 ± 0.01 g, 40 mesh Na-MMT powder, which was pressed tightly into a sample tube by a hydraulic compactor (10 MPa) for 5 min. The linear swelling height over the whole experimental period at atmospheric pressure condition were obtained.

The hot-rolling tests are carried out with shale samples to study the inhibition ability of SX-1, and the retention percentage will be measured after hot-rolling the samples for a specific amount of time at the set temperature. In the experiments, we selected the most representative copolymer shale inhibitors FA-367, KPAM, SIAT and Ultrahib, respectively, and compared the inhibition performance of SX-1. In this work, the obtained (50 ± 0.01 g) 40-mesh shale samples were dried at 105 °C for 24 h, and rolling the shale samples in high-temperature aging tanks filled with 350 mL inhibitor solutions at 120 °C for 16 h. When these experiments were completed, the shale samples was carefully rinsed and filtered by a 100-mesh sieve, then dried to constant weight at 100 °C, weighed, and from this we calculated the hot rolling recovery rate by Equation (1).
(1)$$V_{H}=\frac{R_{t}}{50}\times 100\%$$
where *R_t_* is the recycling mass, *V_H_* is the rolling recovery rate.

#### 2.4.3. Inhibition Mechanism Study

FT-IR was used to study the microscopic binding force between SX-1 and shale. Two grams of SX-1 were added into 100 mL of water-based drilling fluids and stirred for 2 h, then centrifuged for 30 min and the precipitate collected. The precipitate was dried in a vacuum oven to constant weight at 100 °C for 24 h, and then tested via Fourier transform infrared spectrometer in the wave number range of 4000–500 cm^−1^.

SX-1 at certain concentrations was added into WBDFs and stirred for 2 h. Particle size analysis was then performed using a laser diffraction technique (HORIBA, Kyoto, Japan). The operating temperature was 25 °C and the circulation speed was 2000 r/min. The repeatability error was less than 3%, the accuracy error was less than 3%, the frequency of the ultrasonic system was 40 Hz, 70 W, and the stirring speed was 100–475 rpm.

Oriented samples for the XRD analysis were prepared as described in the FT-IR section. Both the dry and wet precipitations were performed on an X Pert PRO MPD diffractometer (PANalytical B.V., Amsterdam, the Netherlands), with the test angle varied from 3° to 20°, and an angle accuracy of 0.0001.

SEM was investigated with an FEI Quanta 450 instrument (Thermo Fisher Scientific, Waltham, MA, USA); the range of the magnification was from 500 to 5000; samples were trimmed from the bottom of the API filter cake; and images were obtained at 120~500 Pa. The acceleration voltage of the FEI Quanta 450 device was 0.02–30 kV, and the probe current was 12 pA–20 nA.

## 3. Results and Discussion

### 3.1. IR Characterization of SX-1

FT-IR spectra were recorded using a WQF-520 Fourier transform infrared spectrometer over a wavenumber range of 4000–400 cm^−1^. Figure 3 shows the FT-IR spectra of AM, AMPS, THAAB and copolymer SX-1. 

The primary amide NH_2_ asymmetric stretching occurred at 3353 cm^−1^ (SX-1, THAAB) and the symmetric stretching band of NH_2_ appeared at 3159 cm^−1^ (AM). The stretching C–H vibrations are present in the region 2970–2850 cm^−1^ [22]. The bending vibrations of water molecules present in the AMPS and in KBr were identified in the shift range of 1690–1630 cm^−1^. The sulfonate group gives the monomer a high degree of hydrophilicity and anionic character over a wide pH range. In addition, AMPS absorbs water readily and also imparts enhanced water absorption and transport characteristics to polymers. The high water-absorbing and swelling capacity when AMPS is introduced to a hydrogel could make it useful in medical applications. 

The vibrational modes of the amide groups may be affected by result of hydrogen bonding. The secondary amide C=O stretching (CONH_2_) was therefore assigned to a peak between 1620–1590 cm^−1^. Four significant peaks for carboxylate groups were observed at 1580–1350 cm^−1^ as a result of the stretching of acrylate. The C–O stretching vibration occurred at 1219 cm^−1^ (AMPS), and in-plane bending vibrations were also found around 1300–1000 cm^−1^, and out-of-plane bending at 1025 cm^−1^, for copolymer SX-1 [23]. The FT-IR result demonstrated that AM, AMPS and THAAB had been successfully copolymerized to form SX-1.

#### 3.1.1. Molecular Weight Measurement

We used a GPC instrument (Alliance e2695) to determine the molecular weight of SX-1, and the result of GPC is shown in Figure 4. *M*_w_ of SX-1 was approximately 13,683 g/mol. With a wide molecular weight distribution, the lower molecular weight SX-1 molecules (800–5000 g/mol) could enter the tetrahedral layer and compress the diffuse electric double layer of shale [24]. In addition, the higher molecular weight end of the distribution could wrap around the surface of the shale.

#### 3.1.2. TGA-DSC Measurement

Thermogravimetric (TGA) differential scanning calorimetry (DSC) was utilized to investigate the thermal stability of SX-1. The thermal gravimetric curve displayed four stages of weight loss, which are shown in Figure 5. The first stage—with a mass loss of 12.8 mass % in the temperature range 40–225 °C—was due to the evaporation of intramolecular and intermolecular water, which was strongly hydrogen-bonded with oxygen atoms and amide groups of SX-1. The second stage took place in the temperature range 225–341 °C with a loss of 44.5 mass (%), and was probably mainly due to the decomposition of amide and quaternary ammonium groups. The third one occurred in the 341–475 °C temperature range with a prodigious loss of 30.6 mass (%), which can be mainly ascribed to the decomposition of SX-1, due to the degradation of C–C bonds in the main chain [25]. 

The final stage, in the temperature range 475–800 °C, showed the main structure of SX-1 to be mostly destroyed. From the DSC curve we noticed a clear heat absorption at 225 °C, which could be attributed to the initial extensive decomposition of SX-1. The combined results of TGA and DSC demonstrate that SX-1 possesses excellent thermal stability. SX-1 therefore has potential application in high temperature and pressure (HTHP) wells.

### 3.2. Inhibition Property Evaluation

#### 3.2.1. Hot-Rolling Recovery Tests

Hot rolling recovery tests are the most basic method for evaluating the performance of a shale inhibitor. If an inhibitor can firmly wrap around the shale surface, it can effectively prevent the hydration and dispersion of shale. The hot-rolling recovery tests were carried out by a roller heating furnace with Longmaxi shale for 16 h at 120 °C, as shown in Figure 6. 

The recovery rate of shale in distilled water was 69.16%, indicating the comparatively high degree of dispersion of Longmaxi shale. The recovery rates of shale increased as the inhibitor solution concentrations were gradually increased. For each concentration, KPAM, Ultrahib and SX-1 demonstrated excellent shale hydration inhibition properties.

The polymer molecules contain functional groups suitable for adsorption, and clay particles can stick to the polymer main chain and will not be hydrated further. SX-1 contains a large number of hydroxyl functional groups. These groups can form hydrogen bonds with the exposed oxygen atoms (O) or hydroxyl groups (OH) on the clay surface. The highest recovery rate values were obtained with an SX-1 concentration of 2.0 wt %, with the recovery rate reaching 90.31%.

#### 3.2.2. Linear Swelling Tests

In this part, linear swelling tests of an Na-MMT artificial core exposed to distilled water and aqueous solutions of different shale inhibitors were studied at room temperature, as shown in Figure 7. 

In all cases, the artificial core samples swelled quickly during the first 100 min, followed by a relatively small expansion over the rest of the test period. The total swelling height of the artificial core, measured over the whole experimental period, was significantly higher in distilled water than in any of the inhibitor solutions. KCl is a well-known and widely preferred inhibitor in drilling fluids. It is generally believed that the low hydration energy and small ionic radius of K^+^ ensure its embedding into the clay. SIAT and Ultrahib are able to enter the tetrahedral layer and compress the diffuse electric double layer of shale due to their small volume [5,7,8]. 

However, the expansion height of the sample in FA-367 and KPAM solution were higher than that in polyamine (SIAT and Ultrahib) and SX-1. The reasons may be that the molecular weight of the two were very large, which would lessen its ability to compress the diffuse double electric layer of the clay. The linear expansion height of Na-MMT artificial core in 2.0 wt % SX-1 solution was just 4.74 mm after 16 h, which was much lower than for the other inhibitors studied.

### 3.3. Inhibition Mechanism Analysis

#### 3.3.1. FT-IR Analysis

FT-IR spectroscopy is a sensitive technique for analyzing the interaction type, configuration, and local environment of Na-MMT modified by an intercalating agent. The adsorption of SX-1 on Na-MMT was confirmed by FT-IR spectra in this work, as shown in Figure 8 below. 

FT-IR of pure Na-MMT shows absorption bands assigned to the characteristic groups of the montmorillonite layer (Figure 8, black line). The stretching vibrations of the structural OH groups occur at 3620 cm^−1^ while the AlAlOH and AlMgOH bending bands are observed at 908 cm^−1^ and 845 cm^−1^, respectively. A strong complex band at 1025 cm^−1^ is attributed to the stretching vibration of the SiO groups of the tetrahedral sheets, while the absorption bands at 525 cm^−1^ and 467 cm^−1^ belong to Si–O–Al and Si–O–Si bending vibrations, respectively. The bands at 3455 cm^−1^ and 1646 cm^−1^ correspond to the OH stretching and bending vibrations of water molecules present in the montmorillonite and in KBr. Absorption bands in the range 800–600 cm^−1^ are assigned to Si–O stretching of quartz and silica [26].

For copolymer SX-1 (Figure 8, red), the characteristic bands include the primary amide NH_2_ asymmetric stretching at 3353 cm^−1^, and the symmetric stretching band of NH_2_ at 3159 cm^−1^ [22,27]. In the spectrum of the Na-MMT/SX-1 composite material (Figure 8, blue) there occurs an absorption band at 1618 cm^−1^, assigned to the primary amide C=O stretching (CONH_2_), while the absorption band at 1400 cm^−1^ corresponds to COO– stretching, which suggests the successful incorporation of SX-1 into Na-MMT [28,29,30]. There are quaternary amine and amide as well as alcohol functionalities in SX-1; adsorption is expected to take place based on the ion-dipole interactions between the polar alcohol groups and the exchangeable cations of shale, as well as H-bonding, van der Waals interactions and entropy effects [31].

#### 3.3.2. Particle Distribution Tests

It is well known that shale possesses a negatively-charged surface which is generally counterbalanced by the positive charge of cations, and flocculation happens when a zwitterionic polymer is added to the drilling fluids. The inhibition performance was therefore further evaluated by particle distribution tests. As shown in Figure 9, the particle size of Na-MMT in distilled water was tiny, only 8.05 μm.

As we all know, when clay is wrapped by polymer, clay dispersion caused by hydration will be prevented, thus making the clay particle size larger. In our experiments, the particle size of Na-MMT increased to 526 μm, 603 μm and 626 μm in 1.5 wt %, 2.0 wt % and 2.5 wt % SX-1 solutions, respectively.

We suggest the following reasons for this: firstly, SX-1 adsorbs onto Na-MMT due to hydrogen bonding and Van der Waals forces, resulting in the negative charge on the surface of Na-MMT being neutralized by SX-1. This compresses the diffuse electric double layer and reduces the repulsion between Na-MMT particles, resulting in the aggregation of Na-MMT. Secondly, SX-1 wraps around the surface of Na-MMT, so that the agglomerated clay no longer disperses.

#### 3.3.3. XRD Analysis

It is well known that clay crystal layers, especially *d*_(001)_, are affected by water molecules, cationic compounds, adsorbents, intercalating agents, and other factors [32]. Therefore, changes of intercalated material between the clay layers can be examined by measuring the change in interlayer spacing, as shown in Figure 10.

XRD patterns shows the dry and wet X-ray diffraction patterns for the *d*_(001)_ diffraction peaks of Na-MMT and Na-MMT/SX-1. As shown in Figure 10a, it can be seen that the interlayer spacing of dry Na-MMT is 1.28 nm, and those of Na-MMT treated with 1.0 wt % and 2.0 wt % SX-1 are 1.44 nm and 1.46 nm, respectively. The diffraction bands are shifted to a lower diffraction angle, corresponding to a higher interlayer distance, indicating the entrance of copolymer SX-1 into the Na-MMT crystal layer. For the wet composite (Figure 10b), due to the hydration expansion of Na-MMT, the spacing distance of wet clay has increased from 1.28 to 1.94 nm compared with that of dry clay, indicating that the interlayer ion was sodium ion [33]. After treatment with SX-1, it was found that the angle shifted to a higher angle, and the interlayer spacing decreased greatly. The combined effect of water outflow and SX-1 entry was the reduction of interlayer spacing. The results showed that SX-1 could enter the internal layer and inhibit the hydration swelling of Na-MMT.

#### 3.3.4. SEM Observations

SEM is a useful technology for observing the morphological changes of Na-MMT composites, and the obtained SEM images are displayed in Figure 11.

Noticeable morphologic differences may be observed in surfaces of the mud cake formed with the drilling fluids. The SEM image of the basic mud cake showed distinct wave-like and rough, homogeneously fluffy-shaped particles as showed in Figure 11a. After treatment with 2 wt % SX-1, the surface of the mud cake becomes smooth and compact; the rough particles were stacked together in an excursive pattern to form slightly curled and crumpled edges—see Figure 11b, which shows the result of reacting SX-1with Na-MMT.

In addition, we also observed the appearance of micro-voids, which is due to the decrease in the strength of interaction between clay sheets. The bigger particle size and holes indicated that intercalation was accompanied by adsorption of SX-1 onto Na-MMT [34,35].

On the basis of our evaluation of the inhibition properties of copolymer SX-1 by a range of methods, as well as analyses of swelling inhibition mechanism reported by other work, we can propose a mechanism of SX-1 for inhibition of Na-MMT hydration. Owing to its hydroxyl and quaternary ammonium functional groups and low molecular weight, SX-1 can adsorb onto clay, enter the inner layer and replace water molecules in the clay crystal layer. It leads to the disappearance of the diffuse double layer; this results in sedimentation occurring (Figure 12a). In the hot-rolling recovery tests, we found that SX-1 showed excellent hydration inhibition properties.

A possible explanation for the phenomenon might be that the larger molecular weight part of SX-1 can be wrapped onto the clay surface. (Figure 12b). [36]. Furthermore, dispersion could not occur as easily with the long SX-1 chains attaching to multiple clay particles, binding them together (Figure 12c).

## 4. Conclusions

In conclusion: SX-1, a novel low molecular weight zwitterionic copolymer for improved wellbore stability was successfully prepared from THAAB, AMPS and AM. The copolymer was characterized by FT-IR, GPC and TGA-DSC. Results showed that SX-1 had a molecular weight of approximately 13,683 g/mol and showed excellent temperature resistance. Its inhibition performance and mechanism were systematically investigated by a range of methods, with experiments indicating that SX-1 possesses superior inhibition properties compared to selected other inhibitors. The analysis of the inhibition mechanism attributed the excellent inhibition properties of SX-1 to three key factors: the multiple driven forces, the wide molecular weight distribution, and the long polymer chain. SX-1 adsorbs strongly on shale due to its the hydroxyl, amine and mono-quaternary amine functional groups. All these features indicate that SX-1 has potential as a shale inhibitor for ensuring wellbore stability in drilling engineering.

## Figures and Tables

**Figure 1 polymers-12-00708-f001:**
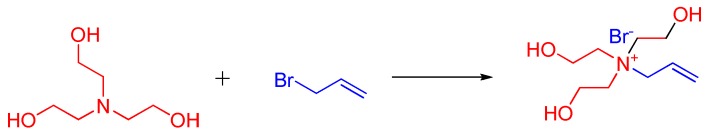
Synthesis route of tris hydroxyethyl allyl ammonium bromide (THAAB).

**Figure 2 polymers-12-00708-f002:**
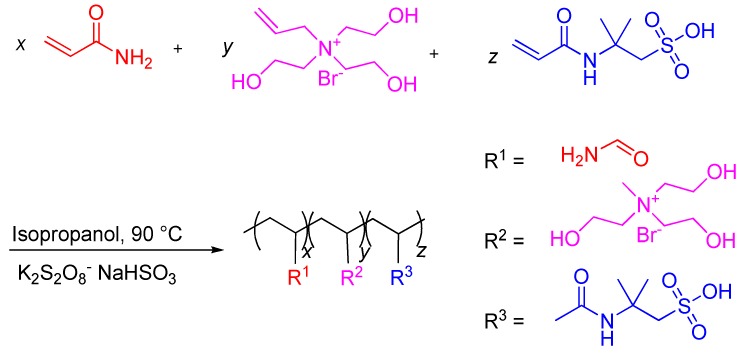
Synthesis route of SX-1.

**Figure 3 polymers-12-00708-f003:**
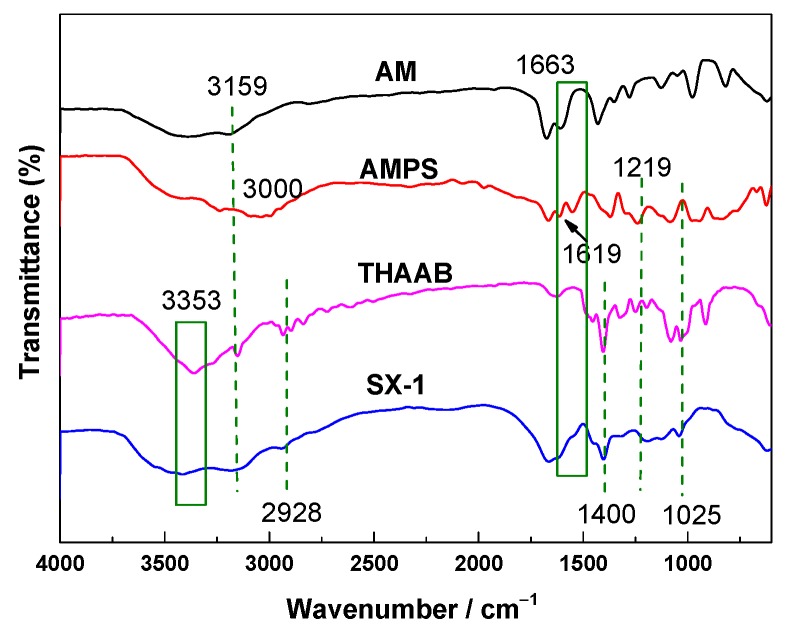
The FT-IR spectra of acrylamide (AM), 2-acrylamido-2-methyl propane sulfonic acid (AMPS), THAAB and SX-1.

**Figure 4 polymers-12-00708-f004:**
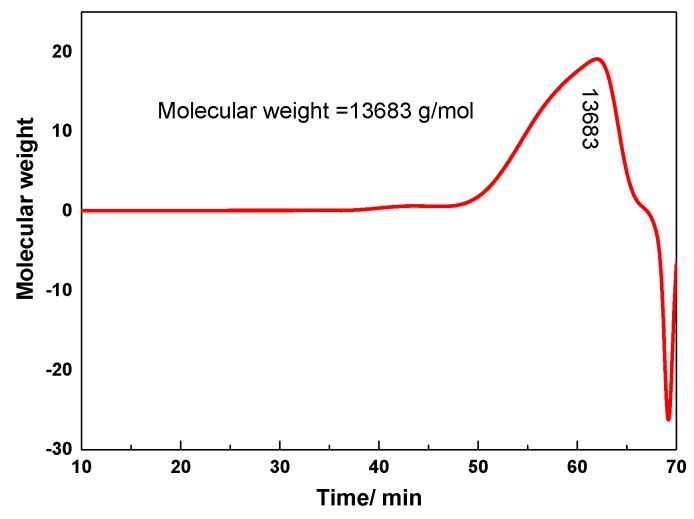
The GPC characterization of SX-1.

**Figure 5 polymers-12-00708-f005:**
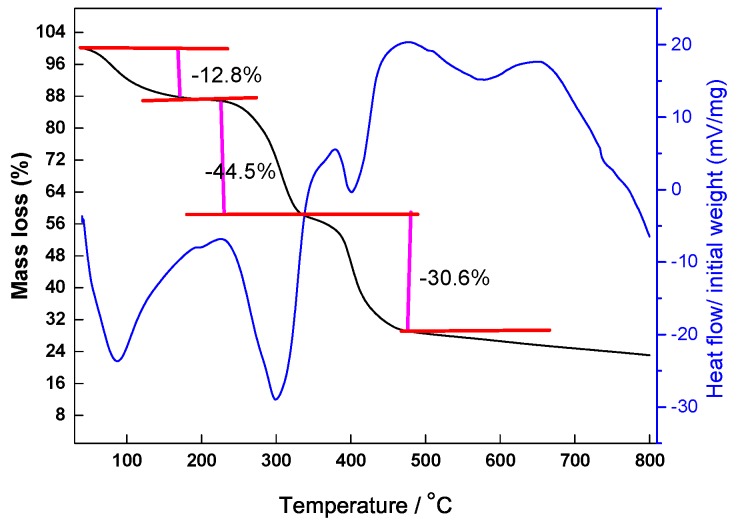
TGA-DSC curves of SX-1.

**Figure 6 polymers-12-00708-f006:**
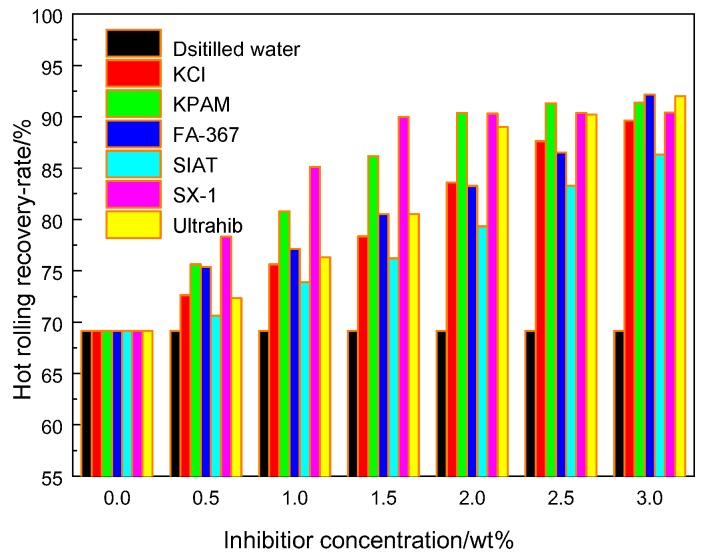
Hot-rolling recovery tests of several shale inhibitors.

**Figure 7 polymers-12-00708-f007:**
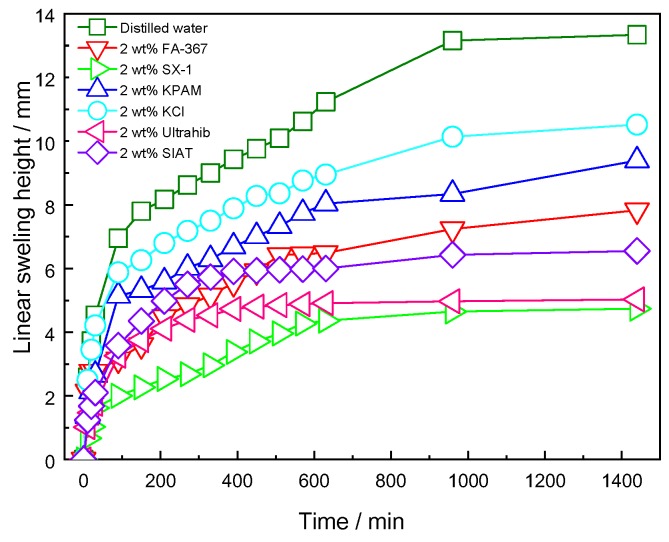
Linear swelling tests of Na-MMT artificial core in different inhibitor solutions.

**Figure 8 polymers-12-00708-f008:**
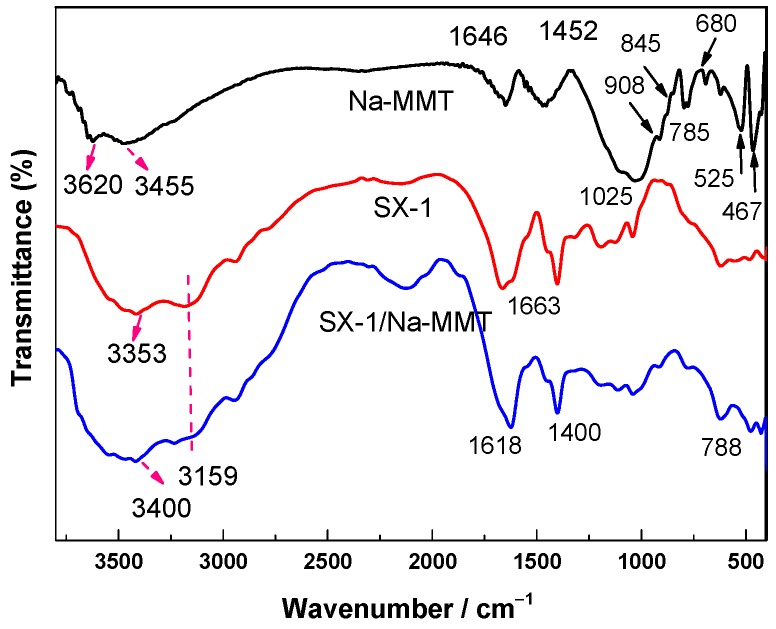
FT-IR analysis of the Na-MMT composites.

**Figure 9 polymers-12-00708-f009:**
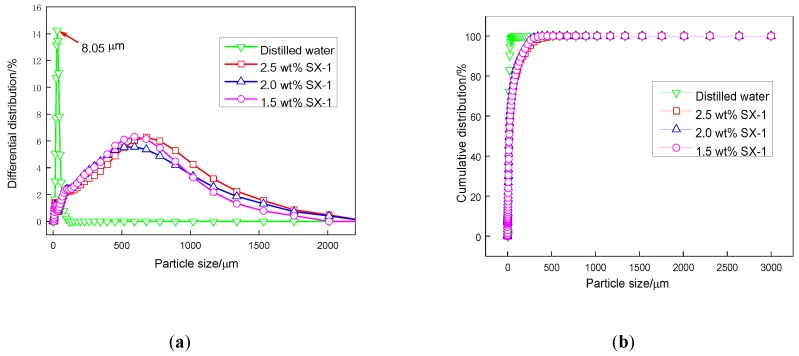
Particle distribution tests of SX-1. (**a**) Differential distribution of samples. (**b**) Cumulative distribution of samples.

**Figure 10 polymers-12-00708-f010:**
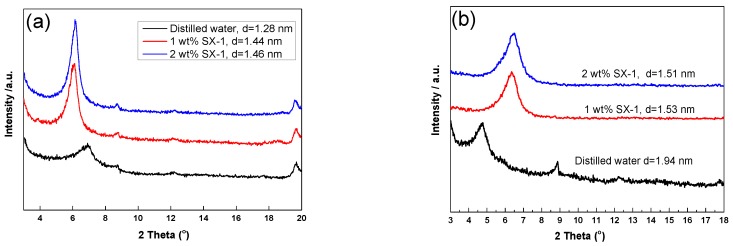
XRD analysis of the Na-MMT composites: (**a**) dry samples, and (**b**) wet samples.

**Figure 11 polymers-12-00708-f011:**
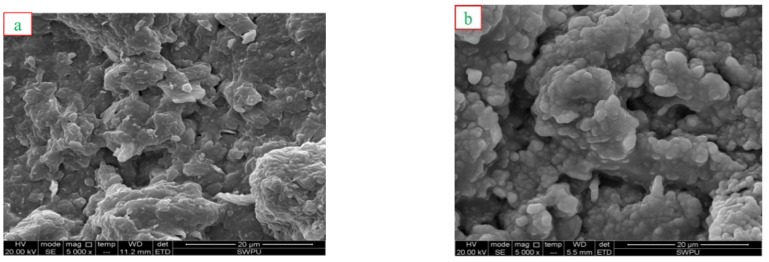
SEM observations of Na-MMT: (**a**) basic mud cake, and (**b**) composite formed after treatment with 2 wt % copolymer SX-1.

**Figure 12 polymers-12-00708-f012:**
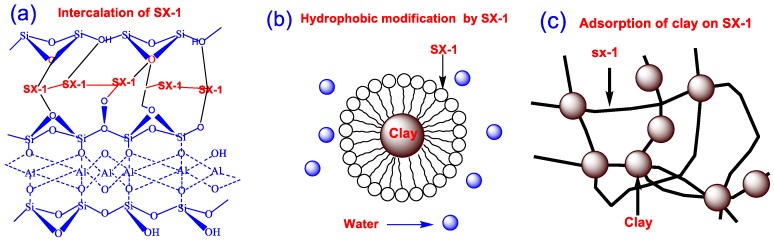
Inhibition mechanism analysis of SX-1. (**a**) Intercalation of SX-1. (**b**) Hydrophobic modification by SX-1. (**c**) Adsorption of clay on SX-1.

**Table 1 polymers-12-00708-t001:** The mineral compositions of shale samples.

Mineral Compositions	Kaolinite	Chlorite	Illite	Sodium Bentonite	Illite/Sodium Bentonite
Content/%	0.0	26.3	65.1	8.6	**10.0**

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
