# Peer review of "The Inhibition Property and Mechanism of a Novel Low Molecular Weight Zwitterionic Copolymer for Improving Wellbore Stability"

_polymers, 2020, doi:10.3390/polym12030708_

Round 1

Reviewer 1 Report

Comments

To line 23 (in Abstract) and to line 259: It is written in the text (lines 159 - 165 and 241 - 254) and shown in Fig. 7 that the linear swelling tests were performed with Na-MMT artificial core, not with shale core.

To line 24 and to lines 242 - 260: The linear swelling tests are characterized by the absolute change in length (mm) of the artificial core without the data of the original core length. It is necessary to represent swelling data accordingly to Equation (1). 

To Figure 9: 1) The correct title for the right figure ordinate is Cumulative distribution (not Calculus distribution). 2) The cumulative distributions for 1.5 wt% and 2 wt% SX-1 must be close to the cumulative distribution for 2.5 wt% SX-1 accordingly to the differential distributions (left figure).

It must be clear stated in the paper text that the majority of tests (swelling, Ft-IR analysis, particle distribution, XRD analysis, SEM observations) were performed for artificial Na-MMT samples. Only hot-rolling recovery tests were performed for the natural shale samples.

Author Response

Thanks to the reviewer for the deep linguistic skills and very professional academic background, we all would like to thank the reviewer for giving me a chance to revise our paper. Next, I will response the comments of Reviewer 1 point by point.

1. To line 23 (in Abstract) and to line 259: It is written in the text (lines 159 - 165 and 241 - 254) and shown in Fig. 7 that the linear swelling tests were performed with Na-MMT artificial core, not with shale core.

Response:  Yes, as pointed out by the reviewer, we use artificial core in the evaluation, and the shale core is a mis-writing of us. We have corrected our errors. Thank you for carefully reviewing of reviewer.

2. To line 24 and to lines 242 - 260: The linear swelling tests are characterized by the absolute change in length (mm) of the artificial core without the data of the original core length. It is necessary to represent swelling data accordingly to Equation.

Response: Thanks to the reviewer’s comments. Both the linear swelling rate and the linear swelling height can be used to study the inhibitive ability of an inhibitor. In this paper, the linear swelling height have been used. In this revise paper, we have deleted the formula of the linear swelling rate and added the calculation formula for the hot-rolling recovery rate.

3. To Figure 9: 1) The correct title for the right figure ordinate is Cumulative distribution (not Calculus distribution). 2) The cumulative distributions for 1.5 wt% and 2 wt% SX-1 must be close to the cumulative distribution for 2.5 wt% SX-1 accordingly to the differential distributions (left figure)..

Response: Thanks to the reviewer’s academic comments, and we have changed it from Calculus distribution to Cumulative distribution. We have verified our experiment data and modified our wrong curve. Thanks for the careful reviewer.

4. It must be clear stated in the paper text that the majority of tests (swelling, Ft-IR analysis, particle distribution, XRD analysis, SEM observations) were performed for artificial Na-MMT samples. Only hot-rolling recovery tests were performed for the natural shale samples.

Response: Thanks to the reviewer’s comments, actually, as the hydration expansion of shale is mainly caused by the expansive of Na-MMT in shale composition, and the evaluation of a shale inhibitor generally by using Na-MMT in the published papers. Longmaxi shale samples are very difficult to obtain and our shale samples are very limited. Therefore, in most of the experiments in this article, we just used Na-MMT instead of natural shale samples. We also acknowledge that there is still a long way to go for field applications of the product. However, we will work hard to do it.

Special thanks to you for your good comments.

Reviewer 2 Report

In this work by Du and coworkers, a novel low molecular weight zwitterionic copolymer was fabricated by copolymerization of tris hydroxyethyl allyl ammonium bromide (THAAB), 2-acrylamido-2- methyl propane sulfonic acid (AMPS) and acrylamide (AM). This process was initiated by a redox initiation system in an aqueous solution. The fabricated copolymer, was characterized by XRD, FT-IR, TGA-DSC, and GPC measurements. Their results show that this novel copolymer show superior properties for ensuring wellbore stability, particularly for high temperature wells. I found this work as an important contribution in the filed, so that I can recommend the publication of this manuscript provided that the authors address the following comments:

1- I recommend a general proofreading and cleanup of the language and grammar. The paper is readable, but sentences are very long and such that it becomes hard to follow the paper.

2-Would be interesting if the authors include more results on the mechanical properties and thermal stability at high temperatures of these novel copolymer.

3- Authors should address how they deal with uncertainties and provide upper and lower bounds. Ideally, they should do an uncertainty analysis as in Advances in Engineering Software, 2016, 100, 19 dealing also with polymers or at least comment on and refer to this contribution.

Author Response

Response to the reviewer 2 comments

Thanks to Reviewer 2 for all your feedback and recommendations. Also, he appreciated the results in our manuscript as very important and beneficial, which we also appreciate.

1- I recommend a general proofreading and cleanup of the language and grammar. The paper is readable, but sentences are very long and such that it becomes hard to follow the paper.

Before submission, the paper underwent a complete review and modification of the language by an English native speaker (native English scientist). However on your recommendation, we have shortened some of the manuscript sentences to make them easier to understand.

2-Would be interesting if the authors include more results on the mechanical properties and thermal stability at high temperatures of these novel copolymer.

In the future, we plan to summarize further results regarding the mechanical properties and thermal stability of this new copolymer at high temperatures. It was not possible to include these results in the current manuscript. Thank you very much for your recommendation, expertise and interest of this manuscript.

3- Authors should address how they deal with uncertainties and provide upper and lower bounds. Ideally, they should do an uncertainty analysis as in Advances in Engineering Software, 2016, 100, 19 dealing also with polymers or at least comment on and refer to this contribution.

We will attempt to include the uncertainty analysis in the following manuscript, where reference is made to the article in Advances in Engineering Software, 2016, 100, 19. This work is very interesting and in the future will come in handy and help us.

Special thanks to you for your care and good comments.

Yours sincerely,

Michal Slaný

Round 2

Reviewer 2 Report

The authors have not addressed my comments. Most importantly, no statistical and uncertainty analysis has been included. As pointed out previously, a simple Matlab code for the UA can be simply taken from Advances in Engineering Software, 2016, 100, 19-31. Without it, I cannot suggest publication of this work as the results are neither reliable nor useful. At least the authors have to include a very detailed discussion on this.

Author Response

Response to the reviewer 2 comments

Thanks to Reviewer 2 for all your feedback and recommendations.

The authors have not addressed my comments. Most importantly, no statistical and uncertainty analysis has been included. As pointed out previously, a simple Matlab code for the UA can be simply taken from Advances in Engineering Software, 2016, 100, 19-31. Without it, I cannot suggest publication of this work as the results are neither reliable nor useful. At least the authors have to include a very detailed discussion on this.

We investigated the uncertainties in a measurement error. In our manuscript, in the introduction section, we have devoted one whole paragraph to this issue. We have included Engineering Software, 2016, 100, 19, as well as other works by these authors on clay polymer nanocomposites in the references section.

Special thanks to you for your care and good comments.

Yours sincerely,

Michal Slaný

Round 3

Reviewer 2 Report

I still think that the authors should do a statistical or uncertainty analysis but I can see that they at least commented on this aspect. Therefore, I suggest publication of this work.